# In-Situ Deposition of Metal Oxides Nanoparticles in Cellulose Derivative and Its Utilization for Wastewater Disinfection

**DOI:** 10.3390/polym12081834

**Published:** 2020-08-16

**Authors:** Mohamed Gouda, Wedad Al-Bokheet, Mohamed Al-Omair

**Affiliations:** Department of Chemistry, College of Science, King Faisal University, Al-Ahsa 31982, Saudi Arabia; snw100@hotmail.com (W.A.-B.); alomair@kfu.edu.sa (M.A.-O.)

**Keywords:** metal oxide nanoparticles, aminated cellulose, in-situ deposition, wastewater disinfection

## Abstract

The target of this work is to investigate and assess the utilization of the synthesized in-situ deposition of metal oxide nanoparticles such as nano-nickel oxide (nNiO), nanocopper oxides (nCuO) and nanoiron oxides (nFe_3_O_4_) in aminated cellulose (Acell), as a protected and compelling antibacterial channel of contamination from domestic wastewater. The prepared Acell and nNiO/Acell, nCuO/Acell and nFe_3_O_4_/Acell nanocomposites were characterized by field emission-scanning electron microscopy (FE-SEM), Fourier transform-infrared spectroscopy (FT-IR), energy-dispersive X-ray spectroscopy, transmission electron microscopy (TEM), selected area diffraction pattern (SAED) and X-ray diffraction techniques (XRD). TEM declared the synthesis of nNiO, nCuO and nFe_3_O_4_ with regular size of 10, 23 and 43 nm, correspondingly. The antibacterial impact of both nNiO/Acell, nCuO/Acell and nFe_3_O_4_/Acell nanocomposites was inspected against Gram-positive microorganisms (*Enterococcus faecalis*, *Staphylococcus aureus*) and Gram-negative microbes (*Escherichia coli*, *Salmonella typhi*) utilizing agar disk diffusion routes. Furthermore, the ability of the synthesized nanocomposites as sterilizers for optional domestic wastewater was studied. The data for the disk diffusion obtained revealed that nFe_3_O_4_/Acell had a greater antibacterial impact than nCuO/Acell and nNiO/Acell. In addition, the purification of domestic wastewater utilizing 1.0 mg of nFe_3_O_4_, nCuO and nNiO in 1 gm of Acell was accomplished by killing 99.6%, 94.5% and 92.0% of total and fecal coliforms inside 10 mins, respectively.

## 1. Introduction

Due to the population growth and industrialization, the world faces a big challenge and serious concerns for suitable clean water. Many of diseases in various countries are most likely linked to the pathogenic microorganisms contaminating the drinking water [1].

The water disinfection process inactivates or kills pathogens prior to human consumption. The most global conventional disinfection processes are categorized into chemical and physical methods, including chlorination [2], ozonation [3] and UV irradiation [4]. On the other hand, previous studies have proven that the formation of disinfection by-products (DBP) are carcinogenic and cannot be avoided globally when chlorine is added to water. Despite the effectiveness of ozonation, it additionally can shape the destructive bromate when ozone responds with bromide particles in water. As compared with the above two methods, UV irradiation is simple, but less vigorous due to it being ineffective in dark water [5]. The drawbacks of these methods encouraged researchers around the world to investigate and apply alternative materials to inactivate water microorganisms. Water treatment experts envisioned that the properties of an ideal disinfectant must be inexpensive, incapable of producing health-threatening DBPs (non-toxic) and an effective disinfectant against a range of many microorganisms under reasonable temperature and pH conditions [6].

For several decades, nanotechnology has presented a new approach for water treatment and become more effective than the conventional technologies. Metals or metals oxide nanoparticles (NPs) have been widely used as antimicrobial agents with extreme toxicity to Gram-positive and Gram-negative bacteria at exceptionally low concentrations, for example, silver [7], titanium dioxide [8] copper oxide, zinc oxide [9] and iron oxide [10]. Some of the metal NPs are expensive and their toxicity to humans and to the environment has been investigated intensively. For example, silver and titanium have been used as metals, either alone or combined with other metals to improve the acts of antimicrobial agents and overcome the cost. Yu Gu et al. showed that the CuFe_2_O_4_ NPs could achieve effective inactivation of *E. coli*, considering the low cost of the materials and a negligible secondary contamination concern [11]. Soheila Asadi et al. synthesized silver-coated Ni_0.5_Zn_0.5_Fe_2_O_4_ and showed that the nanocomposite produced an efficient inhibition of growth against Gram-negative bacteria *Escherichia coli* (*E. coli*) and an efficient elimination of *E. coli*, strengthened through expanding the number of nanoparticles from 2 to 10 g/L, contact time from 10 to 30 mins and a low pH condition [12]. The mechanisms of action of metals NPs for microorganism inactivation depends on the type, dosage and size of nanoparticles and the nature of the target microorganism. The main categories of the mechanisms disinfection is the direct interaction of NPs with the microbial cell membrane and generation of products. For example, reactive oxygen species (ROS) or the release of metal ions causes cell damage [13].

Notably, there is a big concern regarding the presence of NPs in drinking water causing health risks. Some of the metal NPs such as copper and silver cannot be used directly in water because their toxicity is not yet known on humans. They could be dosed at a concentration that does not pose a health concern or they may be impregnated on stabilizing materials such as graphene oxide, chitosan, cellulose or others [14]. Biao Song et al. at loaded Ag NPs on graphene oxide and the results showed the antibacterial activity was more effective against *E. coli* (Gram-negative) than *Staphylococcus aureus* (Gram-positive) bacteria [15]. The high cost of graphene encouraged researchers to use other alternatives as a stabilizing agent of metal NPs that are low cost and environmentally friendly. Chitosan is the second most abundant polysaccharide after cellulose. Mary Helen et al. reported that the nickel–chitosan NPs were active against two types of bacteria: *Staphylococcus aureus* and *Klebsiella pneumonia* [16].

Cellulose is a renewable, almost inexhaustible, raw material and is especially promising as it provides good mechanical performance with potential biocompatibility, biodegradability, non-toxicity and the presence of reactive groups. This last feature is appealing, as their further chemical functionalization could be conveniently used to add a new or modify their properties which will be useful in extending their application to new and unconventional fields [17]. In addition, Gouda et al. synthesized cellulose-graft-polyacrylonitrile copolymer nanofibers containing silver nanoparticles for effective water disinfection. Moreover, they revealed that cellulose-graft-polyacrylonitrile copolymer nanofibers containing AgNPs had great antimicrobial action against *Escherichia coli*, *Salmonella typhi*, and *Staphylococcus aureus*, executing somewhere in the range of 91% and 100% of microbes in a polluted water test and they found that silver nanoparticles were not filtered out from the copolymer nanofibers into the water [18]. Iron and copper nanoparticles were combined with cellulose that acts to solve the problem of aggregation and stability of metal nanoparticles, these nanocomposites provided an excellent result in antibacterial activity.

Furthermore, Alireza et al. worked on the synthesis of cellulose-templated copper nanoparticles and this material inhibited the growth of *Staphylococcus aureus* (Gram-positive) and *E. coli* (Gram-negative) by 95% and 80%, respectively, after 72 h [19]. The diversity of the chemical modification of cellulose creates good properties and new applications, for example, adsorption drug delivery and water disinfection. Aminated cellulose represents one of the cellulose derivatives and it has been successfully achieved by the introduction of amino groups into the active hydroxyl groups of cellulose [20,21].

Nanocellulose is referred to go through a stage for the in-situ deposition of metal oxide nanoparticles, where the different parts of the resultant half-breeds act synergistically towards explicit applications. Be that as it may, commonplace mineralization responses require aqueous conditions or the expansion of further decreasing specialists. Luis et al. [22], show that carboxylated cellulose nanofiber-based films can immediately develop useful metal oxide nanoparticles during the adsorption of substantial metal particles from water, without the requirement for any further chemicals or alterations in temperature. In any case, the in-situ development of metal oxide nanoparticles on cellulose nanocrystals is regularly accomplished by chemical reduction methods [23] or under hydrothermal conditions [24] for which the utilization of poisonous materials and high energy consumption are subjects of concern.

The goal of this work is to deposit, in-situ, nanometal oxides nMO) such as (nNiO, nCuO and nFe_3_O_4_) into aminated cellulose (Acell)) using a co-precipitation method. The prepared nanocomposites will be characterized by Fourier transform-infrared spectroscopy (FT-IR), X-ray diffraction (XRD,) Filed emission scanning electron microscope (FESEM), energy dispersive X-ray spectrometry (EDX) and transmission electron microscope (TEM) techniques. Subsequently, antibacterial activity of the synthesized nanocomposites will be evaluated quantitatively and qualitatively against Gram-positive bacteria such as *Staphylococcus aureus (Staph. Aureus)* and *Enterococcus faecalis* (*E. faecalis)* and Gram-negative bacteria such as *Escherichia coli (E. coli)* and *Salmonella typhi (S. typhi)*. In addition, the purification of domestic wastewater, utilizing the synthesized metal oxide/aminated cellulose (nMO/Acell) nanocomposite, will be examined.

## 2. Materials and Methods

### 2.1. Materials

A native cellulose material with a degree of polymerization (DP) of 10,000, ferric chloride, copper chloride, nickel nitrate, sodium borohydride, isopropanol, H_2_SO_4_ (98%), epichlorohydrine (98%) and ammonium hydroxide (37%) was purchased from Sigma-Aldrich Co., St. Louis, MO, USA. 3-Chloro-2-hydroxypropyl amine was set up as per Ross et al. [25]. Other chemical reagents of analytical grade were also used.

### 2.2. Preparation of Aminated Cellulose (Acell)

Acell was set up as indicated by the revealed strategy [26] as follows: in a stoppered container, native cellulose (1 mole) was completely blended in with sodium hydroxide for 5 mins utilizing a mechanical stirrer. 3-Chloro-2-hydroxypropyl amine was added to the previous blend at 25 °C and completely blended for 5 mins. The prepared blend was kept in a thermostatic water bath for 3 h at 80 °C. Toward the end of the reaction time, the reaction blend was acidified with ethanol containing sulphuric acid and afterwards soxhlet extraction was performed for 12 h utilizing ethanol: water mixture (80:20) and dried at 25 °C. The degree of substitution (DS) of the prepared Acell was determined according to its nitrogen content by Kjeldahl method. The DS of the prepared Acell was 0.33.

### 2.3. In-Situ Deposition of Nanometal Oxides (nMO) into Aminated Cellulose (Acell)

In-situ deposition of nFe_3_O_4_, nCuO and nNiO in the Acell structure were set up as indicated by the described strategy [27]. Acell (0.5 g) was impregnated into a conical flask containing 100 mL of 0.5 M metal salt arrangements at pH 9.5 the blend was continually shaken at 25 °C for 24 h utilizing a bench-top shaker. At the end of the reaction time the prepared metal ion, chelated with Acell, was moved to an ultrasonic bath sonication for 30 mins within the sight of isopropanol/water 50:50 (v/v %). An amount of 0.25 g of sodium borohydride was added to the sonicated blend with consistent mixing using a mechanical stirrer for 30 mins at 25 °C. The prepared nMO/Acell samples were sifted, washed, and then point dried at 60 °C for 1 h. The dried sample was moved into the muffle heater at 300 °C for 24 h.

### 2.4. Characterization

The prepared Acell and nMO/Acell nanocomposites were characterized utilizing field emission scanning electron microscopy combined with an energy dispersive X-ray spectrum. (SEM-EDX; JOELF, Tokyo, Japan). Transmission electron microscope (TEM) was used to investigate the distribution and the average diameter of the prepared nanocomposites (TEM-(ZEISS-EM-10-GERMANY)). The crystal structures of the Acell and nMO/ACell nanocomposites were estimated by means of XRD (Rigaku, Tokyo, Japan) utilizing Cu-Kα radiation (*λ* = 0.154 nm) at 40 kV and 30 mA with a 2θ ranged from 10° to 80°. Moreover, FT-IR spectra were recorded using an FTIR-8400S Spectrometer (SHIMADZU, Kyoto, Japan) to the extent of 500 to 4000 cm^−1^. Total metal fixations inside the Acell were determined according to a reported method [28], utilizing an atomic absorption spectrometer, Varian Spectr A A (220).

### 2.5. Release Study of Metal Oxide Nanoparticles

Triplicate trials were completed in which 1 gm of prepared nMO/Acell was transferred to shaking bottles with 10 mL deionized water. Then the bottles were covered with Parafilm and rotated with a shaker for 48 h at 30 rpm. At the end of the rotation time, the nMO samples were filtered off from the solution using a separation membrane with 0.45 μm pore size. Released nMOs were determined in the filtrate according to the reported method [28]. For every arrangement of metal estimations, an ingestion adjustment bend was built.

### 2.6. Antibacterial Activity Evaluation

Two Gram-positive bacteria (*Enterococcus faecalis* American Type Culture Collection (ATCC) 29,212 and *Staphylococcus aureus* ATCC 6538) and two Gram-negative bacteria (*Escherichia coli* ATCC 11,229 and *Salmonella typhi* ATCC 13311) were utilized for contemplating the tests regarding the antibacterial activity of nMO/Acell. Every single bacterial strain was kept (37 °C for 24 h) in a supplement stock.

#### Disk Diffusion Test Evaluation of the Prepared nMO/Acell Nanocomposites

A correlation among Acell containing different metal oxide nanoparticles was completed utilizing the agar dispersion circle test [29]. An amount of 0.1 mL of each short-term developed bacterial suspension (10^4^–10^5^ CFU/mL) was allowed to feed on the Petri dishes which contained nutrient agar. Circles of both nNiO/Acell, nCuO/Acell and nFe_3_O_4_/AcellNPs were then positioned on the surfaces of Petri dishes. The dishes were incubated at 37 °C for 24 h. The antibacterial activity was qualified dependent on the development of a clear inhibition zone diameter (mm).

### 2.7. Impact of Bacterium–Nanocomposites Interaction Time

Three short-term bacterial suspensions of *Staphylococcus aureus*, *Enterococcus faecalis*, *Escherichia coli*, and *Salmonella typhi* with beginning tallies 3.1 × 10^4^, 3.6 × 10^4^, 4.6 × 10^4^, and 6.4 × 10^3^ a colony forming unit (CFU)/mL were analyzed, respectively. Three examples of each nMO/Acell nanocomposite were immersed into three chambers containing 100 mL of analyzed suspension of bacteria using different interaction times (15 min to 24 h). After each contact time, each bacterial suspension was examined according to the standard methods of water and wastewater examination [28].

### 2.8. nMO/Acell Nanocomposites Utilization as a Wastewater Disinfectant

Local wastewater was utilized as a genuine hotspot for testing the antibacterial effectiveness of prepared nMO/Acell nanocomposites according to the reported method [30]. Bacteria display fecal and total coliforms of untreated and post-treated wastewater were evaluated. Prepared nanocomposite samples were inundated in 200 mL of wastewater for a 10 min interaction period. Fecal and total coliforms were analyzed utilizing the reported method [28].

## 3. Results and Discussion

### 3.1. nMO/Acell Nanocomposites Characterization

#### 3.1.1. FTIR

For recognizing, the various functional groups established a compound in a non-ruinous way for the quantitative and subjective evaluation of biomass parts, Fourier transforms infrared spectroscopy (FTIR) is one of the most generally used techniques. Figure 1A shows the spectra of native cellulose and the assimilation peaks are seen in two wavenumber areas of 3680–2800 cm^−1^ and 1650–400 cm^−1^. The characteristic bands at 3500–3000, 2894, 1428, 1027, 892 cm^−1^ are corresponding to the OH–stretching vibration, C–H stretching vibration, CH_2_ bending vibration, C–O–C pyranose ring vibration and C–O–C stretching of the β(1→4)-glycosidic linkage between the glucose units, respectively [31]. In the case of the primary aminated cellulose spectrum as shown in Figure 1B, there is a similarity with the native cellulose spectrum, in the region which is corresponding to OH stretching vibration there is an increase in the width of the peak which indicates the stretching of the N–H. Moreover, a weak peak appeared with very low intensity at 1300 and 1630 cm^−1^ that indicates the stretching vibrations of the C–N bond and N–H bending vibration, respectively. This is evidence for the presence of the amine substituent on cellulose [32]. After the precipitation of nMO into Acell, there is a difference in the FTIR spectra of the Acell, as shown in Figure 1C–E. The peak that relates to OH becomes low in intensity in nNiO/Acell, as can be seen in Figure 1C, by contrast, it has disappeared in nCuO/Acell and nFe_3_O_4_/Acell, as shown in Figure 1D,E. On the other hand, the peak for C–N appeared in Ni-Acell, Cu-Acell and Fe-Acell with low intensity and weak at 1344, 1374 and 1362 cm^−1^, respectively. The peak for the C–O–C pyranose ring becomes weak and shifted for the all of prepared nanocomposites. Furthermore, the weak characteristic peaks at 457, 481 and 535 cm^−1^ are assigned to nNiO, nCuO and nFe_3_O_4_ vibration, respectively [33,34,35]. In addition to the above, it can be seen that there are a weak peaks at 580, 602 and 436 cm^−1^, which are referred to nNi-N, nCu-N and nFe-N vibration, respectively [36,37]. The interaction between the nMO and Acell is due to two main reasons: the mechanism of self-assembly, which is explained via utilizing hydrogen bonding, and the coordination bonding. For the self-assembly concern, there were two main motivatiors in forming the nMO/Acell, the first was the electrostatic attraction in which the metal atoms in the nMO would self-assemble to form the oxygen atoms in the cellulose matrix [38], while the second driving force was the intermolecular hydrogen bonding, which has been well documented in cellulose-involving materials [39,40]. The coordination bonding concern is the interaction between the nMO and the Acell, which occurred via occupying the lone pair of electrons that exist in the nitrogen of the primary amino group into the partial d orbital of metal ions [41].

#### 3.1.2. XRD

X-ray diffraction is a technique to investigate the crystallinity of materials after modifications, to determine which region is the most crystalline, or amorphous, and the crystalline size of nanoparticles. The crystal size of nanocomposites was estimated by using Scherer’s equation which is represented (1)
(1)$$D_{nm}=\frac{k\lambda }{\beta \cos\theta }$$
where, *D_nm_* is crystallite size (nm), *k* is Scherrer constant = 0.89, *λ* is the X-ray wavelength = 0.15425 nm, *β* is the full width at half the maximum of the peak (FWHM) and *θ* is the Bragg angle [42]. Figure 2A–D illustrates the crystallographic behaviour of the Acell, nNiO/Acell, nCuO/Acell, nFe_3_O_4_/Acell nanocomposite, respectively. According to the previous literature, the crystalline peaks for cellulose were observed around at 2θ of 16 and 22° corresponding to (110) and (200) planes [43]. The crystallinity of cellulose was changed after amination reaction due to the peak at 2θ of 16° was disappeared then a simple shift of the peak at 22° to nearly 25.8°, this indicates the successful formation of aminated cellulose, as shown in Figure 2A. Hence, nMO/Acell nanocomposites show noteworthy changes in Acell pattern. As shown in Figure 2B the existence of nNiO on the surface of the Acell achieved, as shown by the exhibited peaks at 2θ of 44° and 60 × 7° which were identified as peaks of cubic nNiO crystallites, corresponding to crystal planes (200) and (220), respectively. Nevertheless, a peak at 33.5° was identified as Ni_2_O_3_ (004) which, agreed with this finding (JCPDS Card 47–1049) [44]. However, the peak at 25.8° of the crystal Acell did not disappear and this indicated that the nNiO did not change the crystalline structure of the Acell. Furthermore, Figure 2C demonstrates that the nCuO showed considerable variation when combined with the Acell. Strong and sharp peaks appeared at 2θ of 28° (112) for cubic CuO crystallites. There are also three weak peaks at 2θ of 32°, 35.5° and 38.6° which are assigned to the (110), (002) and (111) crystal planes (JCPDS Card 033-0480) [45,46]. The crystal structure of the Acell was completely destroyed, due to the existing peak at 25.8 which shows a good correlation of the copper with the Acell. Lastly, Figure 2D shows the XRD pattern of the nFe_3_O_4_/Acell nanocomposite which is represented by vary weak diffraction peaks at 2θ of 28.5° and 35.5°, these diffraction peaks are characteristic of (220) and (311) planes. Moreover, it can be noted that there is no trace of Acell peak [47]. The calculated average dimeter value of nMO was 9–15 nm. In the meantime, we got a compatible result in the analysis of the selected area electron diffraction (SAED) with XRD data, where the incorporation of the nCuO into the Acell appears to combine better than other metals.

#### 3.1.3. SEM and EDX

Scanning Electron Microscopy is one of the powerful tools used to identify the surface morphology of materials. The microstructure and EDX of the native cellulose, Acell, nNiO/Acell, nCuO/Acell and nFe_3_O_4_/Acell nanocomposites are shown in Figure 3A–D, respectively. The plume shape becomes rough after modification of cellulose with primary amine, as shown in Figure 3A,B. What’s more, there are no agglomerates of nNiO, nCuO and nFe_3_O_4_ and a uniform distribution inside the Acell with a cubic shape can be clearly observed in Figure 4C–D. Alongside this, EDX analysis also confirmed the formation of the nanocomposites and the spectra show the presence of nNiO (signals at 7.8 and 8.4), nCuO (signals at 8 and 9 eV), nFe_3_O_4_ (signals at 6.2 and 7 eV), and no other impurities were detected in the samples spectrum.

#### 3.1.4. TEM & SAED

Transmission electron microscopy (TEM) is a technique which used to analyze the particle’s size diameter and particle distribution of prepared nanocomposites. Figure 4A–C displays the TEM and selected area electron diffraction (SAED) pattern of nNiO/Acell, nCuO/Acell and nFe_3_O_4_/Acell nanocomposites, respectively. TEM images show flat and smooth layers of nanocomposite surfaces without significant aggregation. In addition, it is observed that nMo appears as spherical dark spots, uniformly dispersed inside the Acell. Furthermore, the size distribution of the deposited metals oxide nanoparticles is relatively narrow, and it was found to be about 10, 23, and 25 nm for nNiO/Acell, nCuO/Acell and nFe_3_O_4_/Acell, respectively. Moreover, Figure 4A–C shows the SAED used to characterize the identity and structure of the nMO present on the surface of the aminated cellulose. The area selected for the diffraction pattern with the assigned Miller indices are closely matched to the values obtained by X-ray diffraction. The SAED pictures are proved the crystalline features of the prepared nMO/Acell nanocomposites and the existence of the nMo on the surface of Acell and these nMO crystals assist in scattering the electron beams.

### 3.2. Release of nMO from Acell into Water

It is perceived that nanoparticles may have unfortunate and unanticipated consequences for the environment and in the biological system [48,49]. Along these lines, a water draining test was completed in this examination so as to assess the genuine versatility of nano-metal oxides which can be acquired using a basic one-phase filtering test. The experiment included a mixture of nMO/Acell nanocomposites and deionized water at a ratio of 1:10 (solid: liquid) and disturbing the mixture for 12, 24, and 48 h, then, the solid is filtered and the nMO released in the filtrate is determined. The outcomes produced, indicated that the nMO amount in the filtrate did not surpass 0.01 mg/L after 48 h contact. These outcomes suggest that the antibacterial activity requires close contact of microorganisms with the nMO/Acell nanocomposites, as opposed to being because of the arrival of metal ions in the solution [50,51].

### 3.3. Disk Diffusion Test Evaluation of the Prepared nMO/Acell

The prepared Acell and nNiO/Acell, nCuO/Acell and nFe3O4/Acell nanocomposites were tested for the removal of bacteria using the disk diffusion route. Table 1 shows that the nFe3O4/Acell had a higher antibacterial activity than the nCuO/Acell and nNiO/Acell against both Gram-negative and Gram-positive bacteria and following the decreasing order nFe_3_O_4_/Acell > nCuO/Acell > nNiO/Acell. It may be attributed that the purpose behind the unrivaled antibacterial action of nFe_3_O_4_ is a strong potential partiality towards useful gatherings like amino, carboxyl, or mercapto [52]. The photocatalytic activity of nMO causes photochemical oxidation of intracellular coenzyme-A and consequent alteration of respiratory activities [53,54], rapid leakage of K^+^ ions, and slow leakage of RNA and proteins [55,56], were believed to be the primary cause of cell death and reduced bacterial cell growth. Therefore, photocatalytic activity of nFe_3_O_4_ is higher than that of nCuO, and nCuO is higher than nNiO. This prompts the production of high oxidatin forces that make nFe_3_O_4_ group work such as the carboxyl or amino present in peptidoglycan, which is a part of the cell wall of bacteria. Additionally, Fe^3+^ ions have the ability to eliminate microbes by pulverizing both cell walls and membranes because of their strong reduction ability. Additionally, restricting Fe^3+^ ions to biomolecules may cause protein denaturation, DNA damage and catalyst work substituting [57]. Furthermore, the reasons may be at whatever point nanoparticle goes into the bacterial cell, the death of microbes is expected to be due to the high formation of reactive oxygen species inside the bacterial cell produced during photocatalysis of nMO [58,59]. In Addition, Table 1 shows that the antibacterial action against Gram-positive strains is higher than Gram-negative strains. This is due to the cell wall of gram-negative bacteria is more difficult than gram-positive bacteria which is assumed to be an essential work in the responsibility of passing of nanoparticles into the bacteria cell. In any case, in Gram-negative microorganisms, having Lipoproteins, phospholipids, and so forth, forms a solid boundary that repulses nanoparticles. In another manner, the Gram-positive bacteria have a high negative charge on a superficial level, which assists in the withstanding of nanoparticles that coming from metal nanoparticles, then goes into the cell through spreading utilizing porins as ionic channels and shows strong bactericidal effect [60].

### 3.4. Contact Time Impact of nMO/Acell on to Antibacterial Activity

The impact of contact time of nMO/Acell against *Staphylococcus aureus*, *Enterococcus faecalis*, *Escherichia coli*, and *Salmonella typhi* at various contact times fluctuating from 15 mins to 24 h is shown in Table 2. The data show that, control of Acell without nMO was utilized and had slightly antibacterial effect with reduction rate of percent 7% against all microorganisms due to the presence of the amino group. From Table 2 it is clear that a slow increment in the interaction period enhanced the evacuation effectiveness of the distinctive bacterial strains under scrutiny. A practically measureable decrease in bacteria concentrations was accomplished after 180 mins. The reduction percent increases with an increase in contact time and becomes 99.8%, 99.0%, 96.7% and 96.1% for nFe_3_O_4_/Acell against *Staphylococcus aureus*, *Enterococcus faecalis*, *Escherichia coli*, and *Salmonella typhi* after 24 h, respectively. While, the reduction percent increases with increase in contact time and becomes 94.8%, 94.4%, 92.0% and 91.5% for nCuO/Acell against *Staphylococcus aureus*, *Enterococcus faecalis*, *Escherichia coli*, and *Salmonella typhi* after 24 h, respectively. Furthermore, the reduction percent increases with increase in contact time and becomes 92.5%, 91.1%, 85.0% and 85.0% for nNiO/Acell against *Staphylococcus aureus*, *Enterococcus faecalis*, *Escherichia coli*, and *Salmonella typhi* after 24 h, respectively. The outcomes show that nMO had a compact disinfectant impact on the analyzed bacterial strains. 

### 3.5. Utilization of nMO/Acell Nanocomposites as a Wastewater Disinfectant

Bacterial marker examinations of wastewater before and after they were treated with nNiO/Acell, nCuO/Acell, and nFe_3_O_4_/Acell nanocomposites are shown in Table 3. The data show that the total and feacal coliforms of treated wastewater were decreased after 15 mins treatment with nanocomposites to reach 99.9%. Table 4 shows the effect of treatment time of the nNiO/Acell, nCuO/Acell, and nFe_3_O_4_/Acell nanocomposites on the bacteria. The outcomes display that after 10 mins the total coliforms decreased with an evacuation percent of 99.7% while fecal coliforms diminished by 92%. In any case, an increasing time of 30 mins accomplished practically comprehensive evacuation of total coliforms and fecal coliforms. From this examination, it could be presumed that an exceptionally agreeable sanitization innovation for wastewater can be accomplished utilizing nNiO/Acell, nCuO/Acell, and nFe3O4/Acell nanocomposites with an appropriate contact time. The outcomes are in concurrence with the investigations for the utilization of metal nanoparticles as a disinfectant in wastewater produced particularly from medical clinics containing irresistible microorganisms [61,62,63,64].

## 4. Conclusions

In this work, we studied the synthesis of in-situ depositions of novel nanocomposites based on primary aminated cellulose containing different nanometal oxide such as nCuO, nNiO and NFe_3_O_4_ using a co-precipitation route. The synthesized nanocomposites were confirmed using FE-SEM, FT-IR, TEM, EDX, SAED pattern and XRD spectroscopy. The synthesized nanocomposite was effectively assessed as an antibacterial material for wastewater sanitization against Gram-positive microscopic organisms (*Enterococcus faecalis*, *Staphylococcus aureus*) and Gram-negative microorganisms (*Escherichia coli*, *Salmonella typhi*). The outcomes showed that the bactericidal impact of nano-metal oxides is a promising elective method to the customary compound disinfectants that create harmful purification side effects. Nanometal oxides are stable and are not washed away by a water leaching test after 48 h. In-situ deposition of nanometal oxides into aminated cellulose ends up being a viable antibacterial operator against *Escherichia coli*, *Salmonella typhi*, *Staphylococcus aureus*, and *Enterococcus faecalis* giving sufficient opportunity to infiltrate the multistage cell divider. In any case, after just 10 mins and by utilizing 1.0 mg of nano-metal oxide for every 1 gm of aminated cellulose, over 90% of coliform expulsion was accomplished from rewarded wastewater.

## Figures and Tables

**Figure 1 polymers-12-01834-f001:**
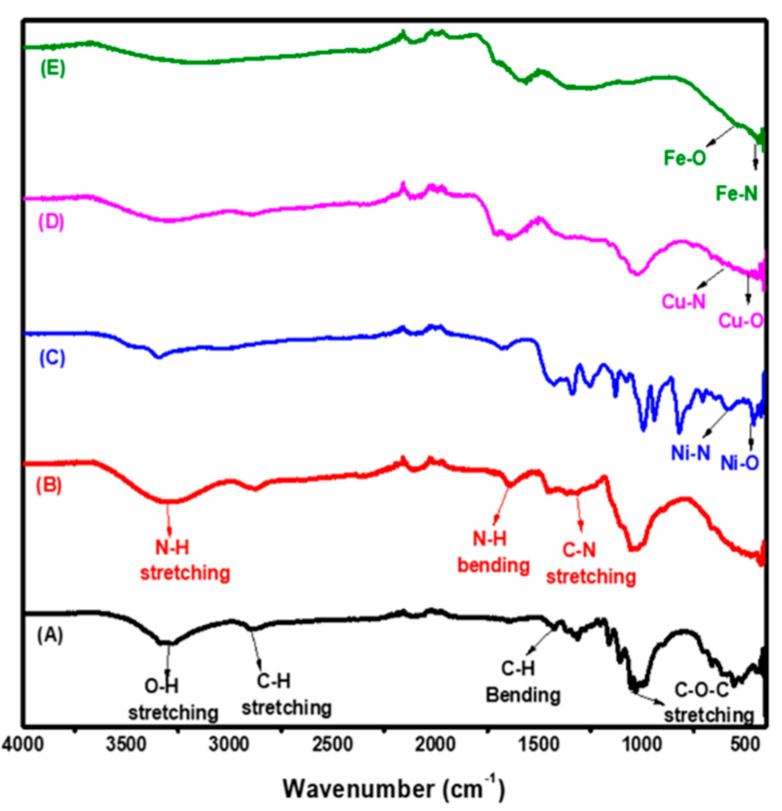
Fourier transform-infrared spectroscopy (FT-IR) spectra for (**A**) Cellulose, (**B**) aminated cellulose (Acell), (**C**) nNiO/Acell, (**D**) nCuO/Acell and (**E**) nFe_3_O_4_/Acell nanocomposite.

**Figure 2 polymers-12-01834-f002:**
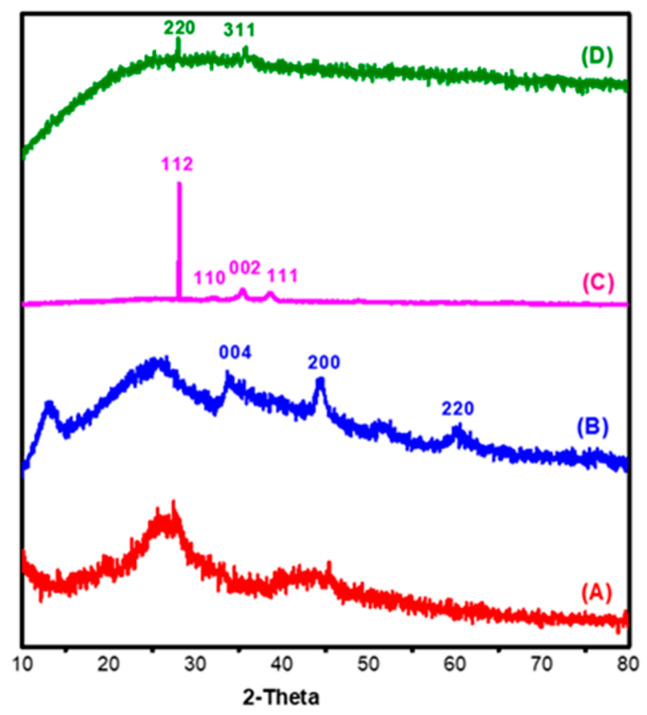
XRD pattern for (**A**) Acell, (**B**) nNiO/Acell, (**C**) nCuO/Acell and (**D**) nFe_3_O_4_/Acell nanocomposite.

**Figure 3 polymers-12-01834-f003:**
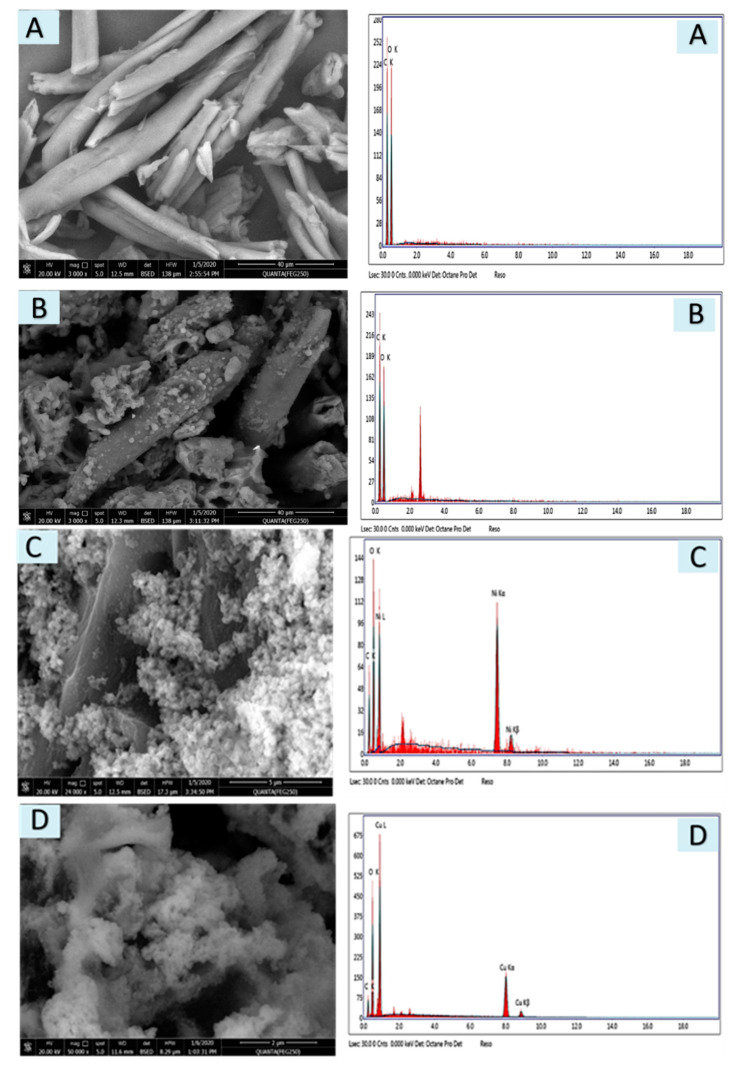
SEM and EDX of (**A**) Cellulose, (**B**) Acell), (**C**) nNiO/Acell, (**D**) nCuO/Acell and (**E**) nFe_3_O_4_/Acell nanocomposites.

**Figure 4 polymers-12-01834-f004:**
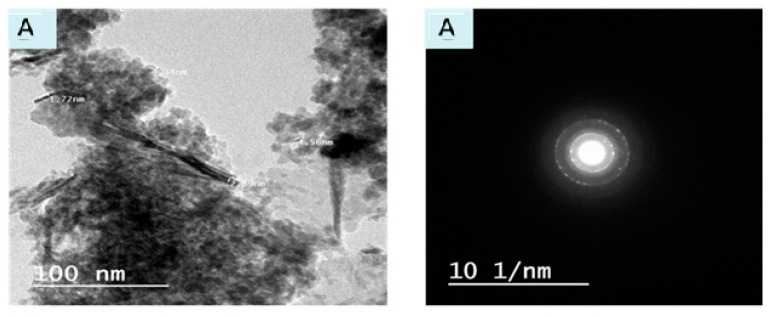
Transmission electron microscopy (TEM) and selected area electron diffraction (SAED) pattern for (**A**) nNiO/Acell, (**B**) nCuO/PAC and (**C**) nFe3O4/Acell nanocomposites.

**Table 1 polymers-12-01834-t001:** Effect of the nMO/Acell type on the microbial removal using agar diffusion disk test.

nMO/Acell Type	Clear Inhibition Zone Diameter (mm)			
	Gram-Negative Bacteria		Gram-Positive Bacteria	
	*Escherichia coli*	*Salmonella typhi*	*Staphylococcus aureus*	*Enterococcus faecalis*
nFe_3_O_4_/Acell	18 mm	16 mm	24 mm	22 mm
nCuO/Acell	12 mm	10 mm	20 mm	17 mm
nNiO/Acell	5 mm	5 mm	10 mm	8 mm

**Table 2 polymers-12-01834-t002:** Effect of contact time of the nMO/Acell nanocomposites on bacterial reduction.

Contact Time (mins)	Bacterial Reduction (R%) and (CFU/100 mL)																							
	nF_3_O_4_/Acell								nCuO/Acell								nNiO/Acell							
	*S. aureus*		*E. faecalis*		*E. coli*		*S. typhi*		*S. aureus*		*E. faecalis*		*E. coli*		*S. typhi*		*S. aureus*		*E. faecalis*		*E. coli*		*S. typhi*	
	CFU	R%	CFU	R%	CFU	R%	CFU	R%	CFU	R%	CFU	R%	CFU	R%	CFU	R%	CFU	R%	CFU	R%	CFU	R%	CFU	R%
* 𝐶_0_	2.1 × 10^4^	32.3	2.5 × 10^4^	30.6	3.6 × 10^4^	21.7	5.3 × 10^3^	17.2	2.3 × 10^4^	25.8	2.8 × 10^4^	22.0	3.7 × 10^4^	19.6	5.5 × 10^3^	14.1	2.5 × 10^4^	19.4	3.0 × 10^4^	16.6	3.9 × 10^4^	15.2	5.6 × 10^3^	12.5
15 mins	1.5 × 10^4^	51.6	1.9 × 10^4^	47.2	3.1 × 10^4^	32.6	4.5 × 10^3^	29.7	1.7 × 10^4^	45.1	2.1 × 10^4^	41.6	3.3 × 10^4^	26.1	4.8 × 10^3^	25.0	1.9 × 10^4^	38.7	2.6 × 10^4^	27.8	3.5 × 10^4^	23.9	5.1 × 10^3^	20.3
30 mins	1.2 × 10^4^	61.3	1.6 × 10^4^	55.6	2.7 × 10^4^	41.3	3.8 × 10^3^	40.6	1.4 × 10^4^	54.8	1.9 × 10^4^	47.2	2.9 × 10^4^	36.9	4.2 × 10^3^	34.4	1.6 × 10^4^	48.4	2.1 × 10^4^	41.7	3.1 × 10^4^	32.6	4.5 × 10^3^	29.7
60 mins	7.2 × 10^3^	76.7	9.7 × 10^3^	73.0	1.6 × 10^4^	65.2	2.6 × 10^3^	59.4	9.7 × 10^3^	68.7	1.3 × 10^4^	63.9	1.8 × 10^4^	60.9	2.9 × 10^3^	54.7	1.3 × 10^4^	58.1	1.6 × 10^4^	55.5	2.1 × 10^4^	54.3	3.2 × 10^3^	50.0
120 mins	3.1 × 10^3^	90.0	4.3 × 10^3^	88.0	9.1 × 10^3^	80.2	1.2 × 10^3^	80.0	6.0 × 10^3^	80.6	8.4 × 10^3^	76.6	1.2 × 10^4^	73.9	1.6 × 10^3^	75.0	7.7 × 10^3^	75.2	1.1 × 10^4^	69.4	1.4 × 10^4^	69.6	1.9 × 10^3^	70.3
180 mins	6.2 × 10^2^	98.0	9.2 × 10^2^	97.0	5.4 × 10^3^	88.2	7.8 × 10^2^	87.8	2.3 × 10^3^	92.5	4.1 × 10^3^	88.6	8.3 × 10^3^	82	1.3 × 10^3^	79.7	3.5 × 10^3^	88.7	5.3 × 10^3^	85.3	1.1 × 10^4^	76.0	1.4 × 10^3^	78.1
1440 mins	6.0 × 10	99.8	3.3 × 10^2^	99.0	1.5 × 10^3^	96.7	2.5 × 10^2^	96.1	1.6 × 10^3^	94.8	2.0 × 10^3^	94.4	3.7 × 10^3^	92	5.4 × 10^2^	91.5	2.3 × 10^3^	92.5	3.2 × 10^3^	91.1	6.9 × 10^3^	85.0	9.6 × 10^2^	85.0

* 𝐶_0_: the concentration of bacterial strain at zero contact time (CFU/100 mL) of *S. aureus* = 3.1 × 10^4^, *E. faecalis* = 3.6 × 10^4^, *E. coli* = 4.6 × 10^4^, *S. typhi* = 6.4 × 10^3^.

**Table 3 polymers-12-01834-t003:** Bacterial markers of crude household wastewater.

Parameters	Raw Wastewater	Treated Effluent		
		nFe_3_O_4_/Acell	nCuO/Acell	nNiO/Acell
Total coliform (MPN-index/100 mL)	2.8 × 10^7^	1.1 × 10^3^	2.2 × 10^3^	3.7 × 10^3^
Fecal coliform (MPN-index/100 mL)	1.5 × 10^7^	2.8 × 10^2^	3.3 × 10^2^	3.1 × 10^3^

**Table 4 polymers-12-01834-t004:** Reduction in total and fecal coliforms of crude household wastewater effluent using nMO/Acell nanocomposites at different contact times.

Time	Control		nFe_3_O_4_/Acell		nCuO/Acell		nNiO/Acell	
	TC	Fc	TC	Fc	TC	Fc	TC	Fc
Zero time	4.1 × 10^3^	1.8 × 10^2^	4.1 × 10^3^	1.8 × 10^2^	4.1 × 10^3^	4.1 × 10^3^	4.1 × 10^3^	4.1 × 10^3^
2 mins	4.6 × 10^3^	1.7 × 10^2^	1.1 × 10^3^	1.4 × 10^2^	2.2 × 10^3^	3.3 × 10^2^	3.7 × 10^3^	3.1 × 10^3^
5 mins	5.1 × 10^3^	2.1 × 10^2^	7.5 × 10^2^	1.1 × 10^2^	1.0 × 10^3^	2.5 × 10^2^	1.8 × 10^3^	4.4 × 10^2^
10 mins	5.7 × 10^3^	2.6 × 10^2^	1.8 × 10	1.5 × 10	6.5 × 10^2^	5.4 × 10	8.2 × 10^2^	7.3 × 10
15 mins	6.2 × 10^3^	3.1 × 10^2^	1.5 × 10	1.0 × 10	2.2 × 10^2^	3.2 × 10	5.3 × 10^2^	5.2 × 10
20 mins	6.4 × 10^3^	4.1 × 10^2^	1.1 × 10	5.3	6.2 × 10	8.8	1.3 × 10^2^	1.1 × 10
30 mins	6.8 × 10^3^	5.8 × 10^2^	1.0 × 10	5	3.1 × 10	6.7	4.7 × 10	9.1

Tc = Total coliforms (MPN-index/100 mL) and Fc = Fecal coliforms (MPN-index/100 mL).

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
