# Peer review of "In-Situ Deposition of Metal Oxides Nanoparticles in Cellulose Derivative and Its Utilization for Wastewater Disinfection"

_polymers, 2020, doi:10.3390/polym12081834_

Round 1

Reviewer 1 Report

1) In the title of this paper, the “In situ deposition” is a key word and innovation of this manuscript. However, in the section of “Abstract” and “Introduction”, the “in situ deposition” is not elucidated. Thus, the authors are suggested to add the content of this point.

2) In order to reduce the risk of nano materials in water, how to separate your metal oxide/ Acell material from water? Are there any residuals in water?

3) What’s the interaction between metal oxide and Acell?

4) The introduction should be expressed in different paragraphs, and each paragraph has one point or subject.

5) In Figure 1, the functional group you analyzed should be added in this figure.

6) The format of reference should be modified with a uniform style, such as Ref.30, the journal name should be lowercase, et al.

7) In author contributions, each author’s name should be added with correspondent contributions.

Author Response

Author's response

< Manuscript ID: polymers-904315>

< In Situ Deposition of Metal Oxides Nanoparticles in Cellulose
Derivative and Its Utilization for Wastewater Disinfection>

We would like to thank the reviewer for his great efforts and for giving useful criticism to the article. Below are the answers to each point.

1) In the title of this paper, the “In situ deposition” is a keyword and innovation of this manuscript. However, in the section of “Abstract” and “Introduction”, the “in situ deposition” is not elucidated. Thus, the authors are suggested to add the content of this point.

In situ deposition is elucidated in the abstract line numbers 13 and 14. Moreover, it was elucidated in the introduction part in the line numbers from 95 to 104.

2) In order to reduce the risk of nanomaterials in water, how to separate your metal oxide/ Acell material from water? Are there any residuals in the water?

First of all, there was no release of nMO from the Acell according to the release study in the manuscript

Second, nMO/Acell can be separated from the water after treatment by filtration

3) What’s the interaction between the metal oxide and Acell?

The interaction between the nMO and Acell is due to two main concerns which, are the mechanism of self-assembly which explained via utilizing hydrogen bonding and the coordination bonding. In the self-assembly concern, there were two main motivations in forming nMO/Acell, the first was an electrostatic attraction in which the metal atoms in the nMO would self-assemble to the oxygen atoms in the cellulose matrix [38], while the second driving force was the intermolecular hydrogen bonding, which has been well known in cellulose-involving materials [39,40]. The coordination bonding concern is the interaction between the nMO and the Acell was occurred via occupying the lone pair of electrons that existing on the nitrogen of primary amino group the partially d orbital of metal ions [41]. Introduced in the manuscript in the lines from 211 to 219.

4) The introduction should be expressed in different paragraphs, and each paragraph has one point or subject.

Done

5) In Figure 1, the functional group you analyzed should be added to this figure.

The functional group was added to Figure 1

6) The format of reference should be modified with a uniform style, such as Ref.30, the journal name should be lowercase, et al.

Done

7) In author contributions, each author’s name should be added with corresponding contributions.

Conceptualization, Gouda, M.; methodology, Gouda, M.; Al-Bokheet, W. A. and Al-Omair, M.; validation, Gouda, M.; Al-Bokheet, W. A. and Al-Omair, M..; formal analysis, Gouda M.; investigation, Gouda, M.; Al-Bokheet, W. A. and Al-Omair, M. ; resources, M. Gouda; data curation, Gouda M..; writing—original draft preparation, Gouda, M. and Al-Bokheet, W. A.;  writing—review and editing, Gouda, M.; and Al-Omair, M..; Gouda, M.; supervision, Gouda, M.; project administration, Gouda, M.; funding acquisition, Gouda, M.

Reviewer 2 Report

The article "In Situ Deposition of Metal Oxides Nanoparticles in Cellulose Derivative and Its Utilization for Wastewater Disinfection" by Mohamed Gouda and co. presents the synthesis in-situ deposition of novel nanocomposites based on primary aminated cellulose containing different nanometal oxide using co-precipitation route and the antibacterial properties of those materials against gram-positive and gram-negative microorganisms.

The paper is well organised and present interesting data, however some minor revisions must be done:

1) From the format of the paper, the caption for Figure 2. XRD\....  it must be just below the figure, with no text between image and caption

2) Table 1, page 10: the effect of nFe3O4/Acell is more pronounced compared to nNiO/Acell (the diameter of inhibiion zone is lhigher for nFe3O4/Acell for all Gram positive and Gram negative bacteria). Is there any explanation for this behaviour? - related to crystaliinity of the samples or to the size of nanoparticles?

Author Response

Author's response

< Manuscript ID: polymers-904315>

< In Situ Deposition of Metal Oxides Nanoparticles in Cellulose
Derivative and Its Utilization for Wastewater Disinfection>

We would like to thank the reviewer for his great efforts and for giving useful criticism to the article.

1) From the format of the paper, the caption for Figure 2. XRD\....  it must be just below the figure, with no text between image and caption

Revised

2) Table 1, page 10: the effect of nFe3O4/Acell is more pronounced compared to nNiO/Acell (the diameter of inhibiion zone is lhigher for nFe3O4/Acell for all Gram positive and Gram negative bacteria). Is there any explanation for this behaviour? - related to crystaliinity of the samples or to the size of nanoparticles?

It may be attributed to that the purpose behind the unrivaled antibacterial action of nFe3O4 maybe it is strong partiality towards useful gatherings like amino, carboxyl, or mercapto [49].

The photocatalytic activity of nMO causes photochemical oxidation of intracellular coenzyme-A and consequent alteration of respiratory activities [43,44], rapid leakage of K+ ions, and slow leakage of RNA and proteins [45,46], were believed to be the primary cause of cell death and reduced bacterial cell growth.

Therefore, the photocatalytic activity of nFe3O4 is higher than that of nCuO and nCuO is higher than nNiO and prompts the production of high oxidative force or authoritative of the nFe3O4 to function group like carboxyl or amino present in peptidoglycan which is a part of the cell wall of bacteria.

Additionally, Fe3+ ions have the ability to eliminate microbes by pulverizing both cell walls and membranes because of their strong reduction ability.

Additionally, restricting Fe3+ ions to biomolecules may cause protein denaturation, DNA harm, and catalyst work substituting [50].

Furthermore, the reasons may be at whatever point nanoparticle goes into the bacterial cell, the death of microbes is expected to either the high formation of reactive oxygen species inside the bacterial cell produced during photocatalysis of nMO [47,48].

Round 2

Reviewer 1 Report

All the problems were considered and revised, and the paper can be accepted.